# Augmenting Radiological Diagnostics with AI for Tuberculosis and COVID-19 Disease Detection: Deep Learning Detection of Chest Radiographs

**DOI:** 10.3390/diagnostics14131334

**Published:** 2024-06-24

**Authors:** Manjur Kolhar, Ahmed M. Al Rajeh, Raisa Nazir Ahmed Kazi

**Affiliations:** 1Department Health Informatics, College of Applied Medical Sciences, King Faisal University, Al-Hofuf 31982, Saudi Arabia; 2College of Applied Medical Sciences, King Faisal University, Al-Hofuf 31982, Saudi Arabia; amalrajeh@kfu.edu.sa (A.M.A.R.); rnahmed@kfu.edu.sa (R.N.A.K.)

**Keywords:** ResNet50, VGG16. augmenting, lung, disease, deep learning

## Abstract

In this research, we introduce a network that can identify pneumonia, COVID-19, and tuberculosis using X-ray images of patients’ chests. The study emphasizes tuberculosis, COVID-19, and healthy lung conditions, discussing how advanced neural networks, like VGG16 and ResNet50, can improve the detection of lung issues from images. To prepare the images for the model’s input requirements, we enhanced them through data augmentation techniques for training purposes. We evaluated the model’s performance by analyzing the precision, recall, and F1 scores across training, validation, and testing datasets. The results show that the ResNet50 model outperformed VGG16 with accuracy and resilience. It displayed superior ROC AUC values in both validation and test scenarios. Particularly impressive were ResNet50’s precision and recall rates, nearing 0.99 for all conditions in the test set. On the hand, VGG16 also performed well during testing—detecting tuberculosis with a precision of 0.99 and a recall of 0.93. Our study highlights the performance of our deep learning method by showcasing the effectiveness of ResNet50 over traditional approaches like VGG16. This progress utilizes methods to enhance classification accuracy by augmenting data and balancing them. This positions our approach as an advancement in using state-of-the-art deep learning applications in imaging. By enhancing the accuracy and reliability of diagnosing ailments such as COVID-19 and tuberculosis, our models have the potential to transform care and treatment strategies, highlighting their role in clinical diagnostics.

## 1. Introduction

Tuberculosis, commonly known as TB, is an illness that primarily targets the lungs and is triggered by a strain of bacteria by the name Mycobacterium tuberculosis. The disease is transmitted through the air when individuals who are infected cough, sneeze or spit. Coronavirus disease (COVID-19) is an infectious disease caused by the SARS-CoV-2 virus. Most people infected with the virus will experience mild to moderate respiratory illness and recover without requiring special treatment. Given the substantial global impact of TB as the leading cause of mortality from infectious diseases before the advent of COVID-19, projections by the Stop TB Partnership suggest that the COVID-19 pandemic could exacerbate TB’s burden [1,2]. Specifically, from 2020 to 2025, an estimated increase of 6.3 million TB cases and 1.4 million TB-related deaths is anticipated [3,4], predominantly affecting the 30 countries with the highest disease burden. Early diagnosis is not just crucial but a matter of utmost importance for both COVID-19 and tuberculosis. Timely detection of these illnesses is crucial for treatment and to stop them from spreading further. Early identification, proactive care, and regular monitoring are not just important [5] but are key to achieving the best possible outcomes in treatment and preventing complications linked with both COVID-19 and tuberculosis [6]. The integration of intelligence (AI) in recent times has significantly advanced the healthcare field, mainly because of the increased access to medical data [7]. These data, comprising an extensive collection of chest X-ray images, have played a role in training AI systems that can accurately spot diseases, predict results, and determine the most suitable treatments [8]. The promise of AI to greatly improve the diagnosis of diseases such as tuberculosis and COVID-19 is a source of optimism, as it can swiftly and precisely analyze images and vast datasets to identify subtle irregularities and assess risk factors, resulting in earlier discovery and treatment. The rapid progress of AI technology has notably expanded the use of AI tools in imaging, significantly improving the assistance these tools offer to healthcare professionals in diagnosing and caring for patients. In our research project, we utilized two learning models, VGG16 and ResNet50, to examine a set of chest X-ray images. We utilized these models to conduct training, validation, and testing for COVID-19 (coronavirus disease), tuberculosis, and cases without any disease (normal condition). Furthermore, we fine-tuned their design to enhance accuracy.

This study presents a deep learning model created for the identification of lung ailments from chest X-ray images with a specific focus on COVID-19 (coronavirus disease), tuberculosis, and cases displaying normal (no disease) conditions. Lung-related diseases pose a public health challenge in the United States, resulting in millions of deaths annually. The lungs are organs that are prone to various illnesses that impact their components. In our investigation, we employed two known deep learning architectures, the VGG16 and ResNet50 models, to examine our collection of chest X-ray images. These models play a role in image recognition and have exhibited exceptional precision in detecting features that are crucial for diagnosing diseases like COVID-19 and tuberculosis as well as conditions indicating a normal lung function.

Section 2 of this paper commences with a review of the literature that places our research within the context of existing knowledge, emphasizing the advancements and prior studies that guided our methodology. Section 3 outlines the approach used to collect and analyze the data, ensuring the transparency and reproducibility of our findings.

In Section 4, we examine the results, delving into the important discoveries and what they mean for the field. Section 5 links these results back to our research goals, offering a profound insight into the studies’ significance. The discussion offers a thorough interpretation of how our findings align with or diverge from the established theories and practices. This analysis helps to clarify the significance of our results within the broader context of the field. Finally, the paper concludes by summarizing the principal conclusions drawn from the research across these sections and suggesting potential directions for future work.

## 2. Literature Review

The study introduces the COVLIAS 1.0 system, a hybrid deep learning model featuring SegNet, VGG-SegNet, and ResNet-SegNet. The model demonstrates superior performance in COVID-19 lung segmentation using CT scans, with significant improvements over the NIH benchmark, achieving high accuracy (AUCs ~0.96 to ~0.98) and rapid processing times (<1 s per image), making it viable for real-time use in clinical settings [9]. The ResNet50-Ensemble Voting model demonstrated excellent accuracy in distinguishing benign from malignant small pulmonary nodules on CT scans, showing great potential for enhancing early-stage lung nodule diagnosis in clinical settings [10]. Ali et al. significantly enhanced lung nodule classification in CT images by employing a decision-level fusion technique with SVM and AdaBoostM2 classifiers and leveraging deep features from advanced architectures, achieving an impressive accuracy of 90.46 ± 0.25% on the LUNGx dataset and outperforming existing methods [11]. The study [12] demonstrated that the BDCNet model, integrating VGG-19 and CNN, significantly outperformed conventional deep learning models in diagnosing COVID-19, pneumonia, and lung cancer from chest radiographs, achieving remarkable classification accuracy and supporting effective early diagnosis. The study [13] revealed that the AlexNet architecture, combined with the SGD optimizer, outperformed other state-of-the-art CNN architectures like LeNet, VGG16, ResNet50, and Inception-V1 in detecting lung cancer on LUNA16 datasets, achieving the highest validation accuracy and overall performance metrics. The study by Dina M. Ibrahim et al. [14]. demonstrated that the VGG19+CNN model excelled in diagnosing COVID-19, pneumonia, and lung cancer from combined chest X-ray and CT images, outperforming other advanced models with exceptionally high accuracy and comprehensive performance metrics across various evaluation criteria. The study [15] highlighted that the integration of Wiener filter preprocessing, GAN segmentation with SSSOA, and CNN classification using VGG16 effectively predicts lung cancer in early stages, achieving an impressive accuracy of 97%, thereby enhancing the potential for early diagnosis and treatment. The study [16] utilized 11 well-known convolutional neural networks to classify COVID-19 and non-COVID-19 lung images, providing critical insights into model performance across varying batch sizes and epochs, thus aiding in the selection of optimal models for enhancing the diagnosis and management of lung-related diseases. Upasana Chutia et al. presented a DenseNet201 model enhanced with CLAHE and a hybrid mechanism that significantly surpasses traditional models in lung disease diagnosis from X-ray images, achieving high accuracy and providing crucial visual insights to support radiological assessments [17]. The study by the authors [18] demonstrated that their proposed deep learning architecture, combining VGG19 with additional CNN layers, excels in classifying various lung diseases, including COVID-19, from CXR images. Achieving an accuracy of 96.48%, a recall of 93.75%, a precision of 97.56%, an F1 score of 95.62%, and an AUC of 99.82%, the model significantly outperformed existing methods, enhancing diagnostic speed and efficiency for healthcare practitioners. Malik, H., Anees, T., Din, M., & Naeem et al. [19], demonstrated that their CDC Net model, utilizing DL techniques, outperformed standard CNN-based models like Vgg-19, ResNet50, and Inception v3 in classifying multiple chest diseases from X-rays, achieving superior accuracy and robustness with an AUC of 0.9953, confirming its efficacy in early and accurate disease diagnosis.

Table 1, below, summarizes the methodologies and key results of various studies focusing on the application of deep learning models for the diagnosis of lung diseases using imaging data: In conclusion, the studies listed in Table 1 explore various deep learning architectures and methodologies to diagnose lung diseases from imaging data effectively. Each model brings advancements in classification accuracy, speed, and diagnostic efficiency, showing potential for real-world clinical application. The results indicate a significant progression towards automating the diagnosis process, thereby enhancing the capability to detect and treat lung diseases at earlier stages.

Our research focuses on enhancing image augmentation techniques to optimize the performance of deep learning models, aimed at improving the diagnosis of significant lung diseases such as COVID-19, tuberculosis, and others. By integrating advanced augmentation strategies, we aim to boost the robustness and generalizability of our models. Future efforts will involve validating these optimized models on unseen data, demonstrating their efficacy in real-world settings, and ensuring they can reliably aid in early and accurate diagnosis across diverse clinical environments. This approach not only pushes forward the boundaries of medical imaging analysis but also promises to enhance diagnostic accuracy, thereby improving patient outcomes in the fight against lung-related illnesses.

## 3. Methodology

In the initial phase of the study, data preprocessing was crucial for the preparation of image data for model ingestion. The dataset was systematically curated into stratified subsets, encompassing training, validation, and testing partitions. The images underwent a resizing process to conform to the input specifications required by the models, which for VGG16 and ResNet50 architectures, is generally set at 224 × 224 pixels. Additionally, an essential preprocessing step involved the normalization of image pixel intensities to a standardized scale, typically normalized to a range between 0 and 1 or standardized to have a mean of 0 and a standard deviation of 1, in accordance with the model’s architectural preconditions. Table 2 outlines how the dataset is distributed across the categories specified by the columns in the table. The table in the image provides information about three types of medical images: tuberculosis, normal, and coronavirus disease. It lists the format, size, and mode of these images. Joint Photographic Experts Group (JPEG) a commonly used method of lossy compression for digital images, particularly for those images produced by digital photography, was used. Tuberculosis is 512 × 512 pixels, which means the image is 512 pixels wide and 512 pixels tall. Normal is 1580 × 1410 pixels, which means the image is 1580 pixels wide and 1410 pixels tall. Coronavirus disease is 2000 × 2000 pixels, which means the image is 2000 pixels wide and 2000 pixels tall.

Additionally, in Figure 1 and Table 3 and Table 4, we employed the VGG16 and ResNet50 models, enhancing the data.

The model summary, provided in the Table 3, describes a neural network architecture based on VGG16 with additional custom layers for classification. The VGG16 functional layer extracts features from input images, resulting in an output shape of (7, 7, 512) with 14,714,688 parameters. The VGG16 functional layer is a pre-trained convolutional neural network that extracts features from input images. It processes the images through multiple convolutional layers, producing a feature map with the shape (7, 7, 512). This means that the network outputs a 7 × 7 grid for each image with 512 feature channels. The layer has 14,714,688 parameters, weights, and biases learned during training. These parameters help the network recognize various features like edges, textures, and objects within the images. The flattening process takes 3D tensor and converts it into a 1D vector. It unrolls all the values in the 7 × 7 × 512 feature map into a single continuous vector. Hence, flattening the 3D output from VGG16 prepares the feature map for the following layers in the neural network, typically fully connected (dense) layers, which require a 1D input. This layer is crucial in learning and mapping the extracted features from the VGG16 output to the final classification or regression task. The dropout layer is a regularization technique used to prevent overfitting in neural networks. It randomly “drops out” (sets to zero) a fraction of the neurons during training, which helps the model generalize better. The VGG16 functional layer is crucial for extracting meaningful features from medical images. It learns from many parameters to help identify patterns such as lesions, tumors, and other anomalies.

Table 4 outlines the layers and parameters of the ResNet50 model, which is also a deep convolutional neural network used for image classification and feature extraction. The model contains many parameters, each contributing to its ability to learn and extract meaningful features from medical images. The Conv2d-1 layer uses 64 filters to scan the input image. Each filter detects specific features in the image, such as edges, textures, or other patterns. Unlike fully connected layers, Conv2d layers are locally connected, meaning each neuron is connected only to a small region of the input image (receptive field), making the layer efficient in capturing spatial hierarchies. The Conv2d-1 layer is foundational in building the network’s ability to understand and analyze images by breaking down the visual data into fundamental components. The BatchNorm2d-2 layer normalizes the inputs for each mini-batch, ensuring that the mean output is close to 0 and the standard deviation is close to 1. Batch normalization reduces the internal covariate shift, making the training process more stable and allowing for higher learning rates. By normalizing the inputs, it helps the model converge faster and can also improve the overall performance and generalization. By introducing non-linearity, the Rectified Linear Unit (ReLU) enables the network to learn and model more complex relationships within the data. Unlike sigmoid or tanh functions, ReLU does not saturate in the positive domain, helping to mitigate the vanishing gradient problem. ReLU is computationally efficient, making it a popular choice for deep neural networks. The Bottleneck-16 layer is a specialized block in the ResNet architecture designed to deepen the network while maintaining computational efficiency. It allows for efficient feature extraction and learning by reducing and then expanding the dimensionality of the feature maps. This layer is fundamental to the ResNet architecture, enabling deep networks to be trained efficiently and effectively by combining dimensionality reduction, convolution operations, and residual connections.

The VGG16 model, known for its simplicity and depth, was characterized by its sequential convolutional layers that effectively captured the textures and patterns in medical images. Using a pre-trained ResNet50 architecture was a strategic choice to classify lung images efficiently. ResNet50 is highly respected for its performance in handling image classification tasks initially created for the extensive ImageNet dataset. It excels at overcoming learning obstacles like vanishing gradients that can impede learning in deep neural networks. We customized ResNet50 to suit our needs precisely. We substituted its layer with a new one tailored to our project requirements. This adjustment maintained the structure of the model, which had undergone training on ImageNet. By keeping the trained layers fixed and concentrating on training the new output layer, we captured the nuances of lung imaging. This approach not only proved effective but efficient in saving resources addressing the unique challenges encountered in medical research environments. By utilizing the knowledge acquired by ResNet50, our model could accurately differentiate between lung conditions. This meticulous adaptation of a known architecture conserved resources and optimized its significant capabilities making it an excellent option, for analyzing medical images.

Furthermore, we applied, augmenting techniques for dataset (images) were applied to artificially expand the variety of data available for training models, particularly when using advanced neural networks like ResNet50 and VGG16, to achieve better results. These procedures involved making systematic modifications to images in a dataset to create new, altered versions of the same images. Common augmentations included rotations, scaling, cropping, flipping, and adjustments to the brightness, contrast, or color saturation of images. The primary purpose of these techniques was to enhance the robustness and generalization capability of image-processing algorithms. By exposing models like ResNet50 and VGG16 to a broader range of variations of the same image, these techniques helped prevent overfitting, where a model performs well on training data but poorly on unseen data.

### 3.1. Dataset

The dataset contains X-ray images of the chest created by merging and organizing datasets to include four types of lung diseases. Additionally, there is a folder dedicated to lung images. To expand the dataset, augmentation was used, resulting in a total of 10,000 images. In our study we specifically examined two diseases—COVID-19 and tuberculosis—and normal lungs. These diseases are separated into folders for training, validation, and testing datasets. For details, please refer to the hyperlink provided below.

https://www.kaggle.com/datasets/omkarmanohardalvi/lungs-disease-dataset-4-types, accessed on 20 November 2023.

### 3.2. Augmentation

Image augmentation and normalization are crucial in familiarizing DL models [20,21,22], especially for image classification. DL models require that all input images be of the same size [23,24]. Resizing and cropping are necessary to ensure that each image fed into the model has the exact dimensions. If the images vary in size, the model will not be able to handle them in batches, which can lead to errors or require more complex preprocessing [25,26,27]. Utilizing methods such as cropping can assist the model in improving its attention on areas of an image while undergoing training, ultimately enhancing its capacity to identify objects regardless of their placement within the image. This results in improved adaptability when the model is presented with images. Standardizing the image data when using models pre-trained on a dataset like ImageNet is essential [28,29]. Many COVID-19 radiography image datasets come in various formats, standards, sizes, and quality levels. These variations pose challenges for researchers looking to advance AI research related to COVID-19. Moving forward, it would be beneficial to establish operating procedures for COVID-19 radiography images [30]. These models are fine-tuned to work with input distributions characterized by values and standard deviations for each color channel. Applying the standardization process ensures that the feature distribution of the input image aligns with that of the dataset used for pre-training, helping the pre-trained model perform effectively. We applied resizing and cropping, then conversion to Totensor, and finally, normalization was applied; refer to the programming code that is supplied with this manuscript.

We used the following evaluation metric for our proposed models. Precision refers to the ratio of correctly predicted positive observations to the total predicted positives. It indicates the accuracy of the positive predictions.
$$\mathrm{Precision}=\frac{True\;Positives\;(TP)}{True\;positive\;(TP)+False\;positive\;(FP)}$$

Recall (Sensitivity) refers to the ratio of correctly predicted positive observations to all observations in the actual class. It measures the model’s ability to capture positive cases.
$$\mathrm{Recall}=\frac{True\;Positives\;(TP)}{True\;positive\;(TP)+False\;negative\;(FN)}$$

The F1-Score is the harmonic means of precision and recall, providing a balance between the two. It is useful when you need to seek a balance between precision and recall.
$$F1\text{-}\mathrm{Score}=2\times \frac{precision\times recall}{precision+Recall}$$

An ROC Curve (Receiver Operating Characteristic Curve) is a graphical representation of a classifier’s performance. It plots the true positive rate (recall) against the false positive rate (1—specificity) at various threshold settings. It shows the trade-off between sensitivity and specificity. The AUC (Area Under the ROC Curve) is a single scalar value to summarize the performance of a classifier. It ranges from 0 to 1, with a value closer to 1 indicating a better-performing model.

## 4. Results and Discussion

Based on the results and evaluation metrics, a radiologist can conclude that the ResNet50 model exhibits outstanding performance in diagnosing COVID-19 and normal lung conditions, with near-perfect precision, recall, and F1-scores, and a high ability to detect tuberculosis, though with some room for improvement. The ROC curves and AUC values confirm the model’s high discriminatory power. The benefits of the model include providing high diagnostic accuracy, reducing misdiagnosis, and ensuring appropriate patient care while preventing unnecessary procedures for healthy patients. It enhances efficiency in clinical settings by providing rapid, consistent diagnoses, which is crucial during high-demand periods like pandemics. It supports radiologists by serving as a second opinion, reducing their cognitive load. The model optimizes resource allocation by accurately identifying cases without disease, improving radiology department workflow and patient management. Additionally, the model can be continuously improved with more data and fine-tuning, providing a platform for training new radiologists in AI-supported diagnostics. Implementing the ResNet50 model in clinical settings can significantly enhance diagnostic accuracy, efficiency, and resource management, benefiting healthcare providers and patients.

In the evaluation of lung imaging using the ResNet50 deep learning model, the classification reports and visual results from various sets demonstrate the model’s excellent diagnostic precision. This model has proved its capability in effectively distinguishing between normal lung scans and those showing signs of tuberculosis and coronavirus disease.

The initial graph as shown in the Figure 2, depicting training and validation accuracy, reveals significant fluctuations in the training accuracy, suggesting variability in the learning process—potentially due to factors like mini-batch selection or adjustments in learning rates. It also illustrates training and validation losses, both tapering off over time yet marked by noticeable volatility. The pronounced decline in training loss, consistently lower than the validation loss, suggests effective learning, as evidenced by the less smooth reduction in validation loss.

Table 5 shows the classification report for the validation set and reveals excellent performance of the model across three categories: coronavirus disease, normal, and tuberculosis. Each category shows high precision, recall, and F1-scores, with values nearly or exactly at 0.97 and 0.99, indicating the model’s robust capability in accurately diagnosing and distinguishing these conditions. Specifically, the model achieves nearly perfect results for ‘normal’ conditions, demonstrating its effectiveness in accurately identifying cases without disease, which is crucial for preventing false diagnoses in clinical settings. For ‘coronavirus disease’ and ‘tuberculosis’, the model also shows high effectiveness with scores around 0.97, underscoring its utility as a reliable diagnostic tool in detecting these significant pulmonary diseases with minimal error rates. Table 5 showcases how well our model performs in identifying COVID-19, normal lung conditions and tuberculosis. The precision and recall rates are impressively high, ranging from 0.98 to 1.00 with F1 scores reflecting these results. This demonstrates the reliability and accuracy of the model. Notably, the flawless recall rate for lung conditions underscores its ability to accurately detect cases without any illness present, which is crucial for avoiding procedures and treatments for patients. The model also shows high performance in diagnosing tuberculosis, with errors highlighting its efficiency in identifying this condition. In summary, our model proves to be a tool with the accuracy and precision that are particularly beneficial in settings where accurate disease detection is essential for optimal patient care and treatment.

The confusion matrices for both training and validation sets illuminate the model’s adeptness at accurately classifying the conditions, as shown in the Figure 3. The left graph displays the confusion matrix from the training dataset, where the model achieves high accuracy in classifying COVID-19, normal, and tuberculosis (TB) cases, with minimal misclassifications. The right graph represents the validation dataset, showing how the model performs with new, unseen data, with slightly higher misclassification rates for TB and COVID-19, indicating areas for improvement in model generalization. This comparison underscores the model’s robust training performance while highlighting the need for further optimization to improve validation accuracy. The confusion matrices illustrate the ResNet50 model’s strong performance in classifying COVID-19, with high true positive rates and low misclassification rates. The model’s reliability and accuracy make it a valuable tool for assisting radiologists in diagnosing COVID-19 from medical images. Continuous monitoring and refinement based on confusion matrix insights can further enhance the model’s clinical utility.

The ROC curve and associated AUC values provide robust evidence that the ResNet50 model performs exceptionally well across all three conditions in the test data, as shown in Figure 4. It shows nearly perfect or perfect discriminatory ability to distinguish between normal conditions, COVID-19, and tuberculosis, which is consistent with the high precision, recall, and F1-scores noted earlier.

Figure 5 shows the ROC curves for the ResNet50 model, assessing its diagnostic accuracy across three classes: coronavirus disease (blue line, AUC = 1.00), normal lung conditions (green line, AUC = 1.00), and tuberculosis (red line, AUC = 1.00). The curves demonstrate the model’s exceptional ability to correctly classify each condition with minimal false positives, highlighting its robustness in clinical diagnostic settings.

The test set confusion matrix shows an outstanding ability of the model to correctly diagnose coronavirus disease, with 399 true positives out of 406, and only eight cases misclassified as tuberculosis, showing a robust sensitivity for this condition, as shown in Figure 6. The model performed flawlessly in identifying all cases without errors, a crucial factor in avoiding false alarms in medical environments. The ROC curves of the Res model showcase its ability to detect lung ailments, achieving perfect AUC scores for COVID-19, normal cases, and tuberculosis instances. This means that the model is highly dependable and efficient, serving as an aid for radiologists in diagnosing these conditions based on images. With its sensitivity and specificity, the model ensures detection and categorization, leading to improved diagnostic precision and effectiveness in clinical practice.

The ResNet50 model is most effective in diagnosing COVID-19 and identifying normal lung conditions, showing excellent performance that slightly improves or maintains in validation. However, the model needs improvement in tuberculosis detection for better generalization to unseen data. Enhancing its ability to detect tuberculosis without sacrificing accuracy for other conditions would make it more robust and reliable for broader clinical use.

The provided figures, bar charts, and confusion matrices illustrate the performance of the VGG16 model in diagnosing lung conditions such as COVID-19, normal lung conditions, and tuberculosis. Furthermore, they have provided a clear and comprehensive understanding of the VGG16 model’s performance, highlighting its strengths and areas for further development. This ensures that it is a reliable diagnostic tool in clinical settings.

Based on the findings and assessment criteria, a radiologist can determine that the ResNet50 model shows performance in diagnosing COVID-19 and normal lung conditions. It demonstrates near precision, recall, and F1 scores, along with an ability to detect tuberculosis, though there is room for enhancement. The ROC curves and AUC values validate the model’s capability. The advantages of this model include diagnostics, reducing misdiagnoses for patient care, and avoiding unnecessary procedures for healthy individuals. It streamlines processes by delivering consistent diagnoses, especially during times of high demand, like pandemics. Furthermore, it helps radiologists ease their workload. The model optimizes resource allocation by pinpointing disease cases and improving workflow in radiology departments and patient care. Moreover, continuous refinement with data and adjustments offers a platform for training radiologists in AI-assisted diagnostics. Integration of the model in clinical environments can significantly boost diagnostic precision, operational efficiency, and resource utilization to benefit both healthcare providers and patients.

Table 6 shows the classification reports for the VGG16 model on both validation and test datasets, demonstrating its effectiveness in diagnosing three specific conditions: coronavirus disease, normal lung conditions, and tuberculosis. In the validation dataset, the model shows a commendable performance with high precision and recall, particularly excelling in identifying normal conditions, with nearly perfect scores (0.99 for both precision and recall). However, our model does show some room for improvement in identifying tuberculosis, as indicated by a slightly lower recall rate of 0.89. This suggests that while the model is highly effective, it can sometimes miss cases of tuberculosis. In contrast, when tested on a new dataset, the model demonstrates enhanced performance, showing higher precision and recall across all conditions. Specifically, for COVID-19, the model achieves an impressive precision of 0.92 and a recall of 0.98, maintaining high accuracy for normal lung conditions as well. Our model has made significant progress in detecting tuberculosis, with precision reaching 0.99 and recall increasing to 0.93. These advancements demonstrate the model’s improved capacity to identify tuberculosis in datasets accurately. In general, these findings emphasize the capabilities of the VGG16 model, especially its ability to apply knowledge gained from validation scenarios to real-world testing situations. Hence, our model underscores its potential utility in clinical settings, where the model’s accurate and reliable disease identification can be crucial for effective patient management and treatment planning. The VGG16-based model demonstrates high accuracy and robustness in classifying medical images for coronavirus disease, normal, and tuberculosis. The high precision and recall values indicate that the model can consistently distinguish between disease and normal cases, making it an effective tool in medical diagnostics. For coronavirus disease and normal cases, the model shows exceptional performance. While the model is slightly less precise in detecting tuberculosis during validation, it performs better on the test set, indicating its potential for clinical application in identifying these conditions.

Figure 7 in our report presents the confusion matrix for the validation set, showcasing the VGG16 model’s performance in classifying COVID-19, normal lung conditions, and tuberculosis. The model performs exceptionally well in identifying COVID-19, correctly diagnosing 395 cases, though it does misclassify nine cases as tuberculosis and two as normal. This highlights the model’s strong capability to detect COVID-19, albeit with some confusion when it comes to distinguishing it from tuberculosis. For cases categorized as normal, the precision is very high, indicating that the model is extremely effective at identifying non-diseased conditions with minimal error, though the exact number of true positives is not shown in the figure. However, the issue of misclassifying COVID-19 as tuberculosis is a point of concern, pinpointing an area where the model needs improvement. Enhancing the model’s ability to differentiate more effectively between these two conditions is crucial to prevent potential false diagnoses in a clinical setting. This improvement could lead to more accurate treatment decisions and better patient outcomes.

Figure 8 displays the confusion matrix related to the classification of COVID-19, normal lung conditions, and tuberculosis. The matrix reveals that the model performs highly in identifying COVID-19, as evidenced by 400 true positive cases, showcasing a high level of precision with minimal confusion regarding other conditions. Nonetheless, there are some inaccuracies observed in the model’s performance, such as misclassifying four COVID-19 cases as tuberculosis and three as normal cases. These findings suggest that our model may face challenges in distinguishing between these diseases. On a note, the model demonstrates a capability in recognizing cases classified as normal despite a slight overlap with COVID-19 in certain misclassifications. However, the slight misclassifications between COVID-19 and tuberculosis underscores an area where further refinement is warranted. Improving the model’s accuracy in distinguishing between these diseases could significantly enhance its practicality, in settings leading to decision-making processes and better patient outcomes. The model also demonstrates a strong ability to recognize cases categorized as normal, although there is a slight overlap with COVID-19 in some misclassifications. Despite these minor issues, the overall results affirm the model’s robust diagnostic capabilities for identifying these conditions. The ResNet50 model was enhanced using augmentation methods, demonstrating the ResNet50 model’s accuracy and dependability in identifying COVID-19, normal conditions, and tuberculosis cases. Utilizing this model results in a greater number of diagnoses compared to manual image analysis, leading to better patient outcomes. Integrating this technology into settings enables radiologists to provide precise diagnoses, ultimately improving patient care and optimizing healthcare resources.

Figure 9 depicts the ROC (receiver operating characteristic) curves for a multi-class classifier designed to identify three distinct conditions: COVID-19, normal (healthy state), and tuberculosis. Each curve represents one of the classes, and they all show perfect classification performance, with an area under the curve (AUC) of 1.00. This perfect AUC score means that the classifier is exceptionally effective at distinguishing between affected and unaffected individuals for all the categories. It achieves high levels of both sensitivity (the ability to detect positive cases) and specificity (the ability to avoid false alarms) across the board, without any increase in false positive rates even in the regions where sensitivity is highest. This level of accuracy indicates that the classifier is highly reliable for medical diagnostics, providing confidence in its use for disease detection without misclassifying healthy or diseased individuals.

Figure 10 shows a comparison of AUC (area under the curve) outcomes among cutting edge techniques, including a recently introduced method for identifying lung ailments from chest X-ray images. Each bar on the x axis is tagged with a reference number. The y axis illustrates the range of AUC results, spanning from 0.75 to 1.00. A higher AUC score indicates a higher performance of a model in distinguishing between diseased and healthy states. In this assessment, our proposed method exhibits superiority over numerous existing approaches, underscoring the effectiveness of the novel strategy employed in this research. This strategy harnesses methodologies like data augmentation and balancing to enhance the classification process. ResNet50 stands out for its results, indicating its potential as a choice for assisting in clinical diagnoses of illnesses such as pneumonia, COVID-19, and tuberculosis. It positions the proposed approach as a step in ongoing efforts to improve the accuracy and dependability of diagnosing lung diseases. This advancement aims to contribute to patient care and treatment strategies.

## 5. Conclusions

The VGG16 and ResNet50 models have shown accuracy when identifying lung conditions like tuberculosis, COVID-19 and normal lung states. Overall, ResNet50 tends to perform better than VGG16. These models have proven to be effective thanks to data augmentation, demonstrating their ability to generalize well on data. Their high precision and recall rates highlight their potential for use in diagnostics, marking a significant advancement in automated disease detection through image analysis. While VGG16 initially lagged behind ResNet50 in validation, it showed improvement during testing. In the validation phase, the model achieved precision rates ranging from 0.88 to 0.99 and recall rates from 0.89 to 0.98. During the testing phase, we observed a notable increase in both precision and recall rates. The model achieved a commendable score of 0.99 for detecting tuberculosis, indicating progress in diagnosing this condition. Our future efforts will be dedicated to improving the model’s reliability by expanding its range to cover more conditions and seamlessly integrating them into real-time clinical diagnostic systems. Additionally, our exploration of data integration and advanced data augmentation techniques will further enhance accuracy. However, it is crucial to underline the urgency of addressing biases and ensure that these models perform consistently across diverse populations as they approach clinical use.

Future work aims to enhance the models’ resilience by expanding the range of conditions it can detect and integrating them into real-time diagnostic systems. Exploring data integration and employing data augmentation strategies will also boost diagnostic accuracy. Ensuring performance across populations becomes critical as these models approach clinical implementation.

## Figures and Tables

**Figure 1 diagnostics-14-01334-f001:**
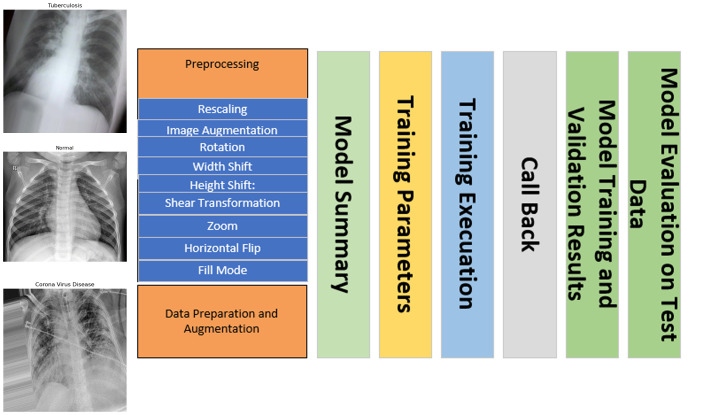
Working model under training and validation along with data preparation and augmentation.

**Figure 2 diagnostics-14-01334-f002:**
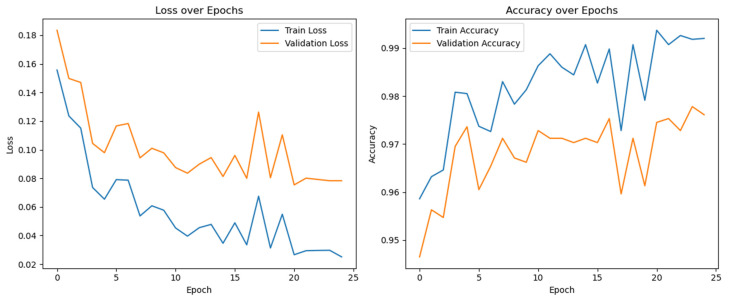
Training and Validation result in terms of loss and accuracy of ResNet50 model.

**Figure 3 diagnostics-14-01334-f003:**
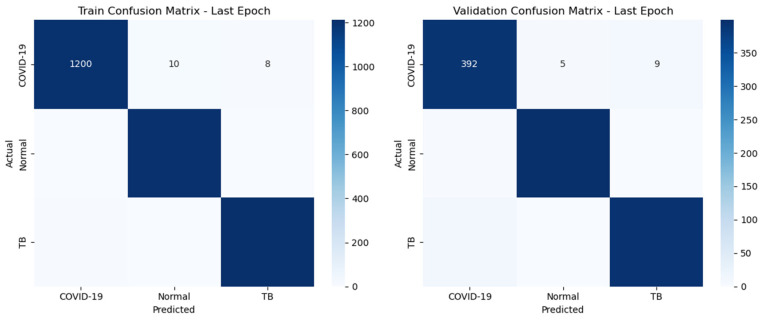
Train and validation confusion matrices for the ResNet50 model at the last epoch.

**Figure 4 diagnostics-14-01334-f004:**
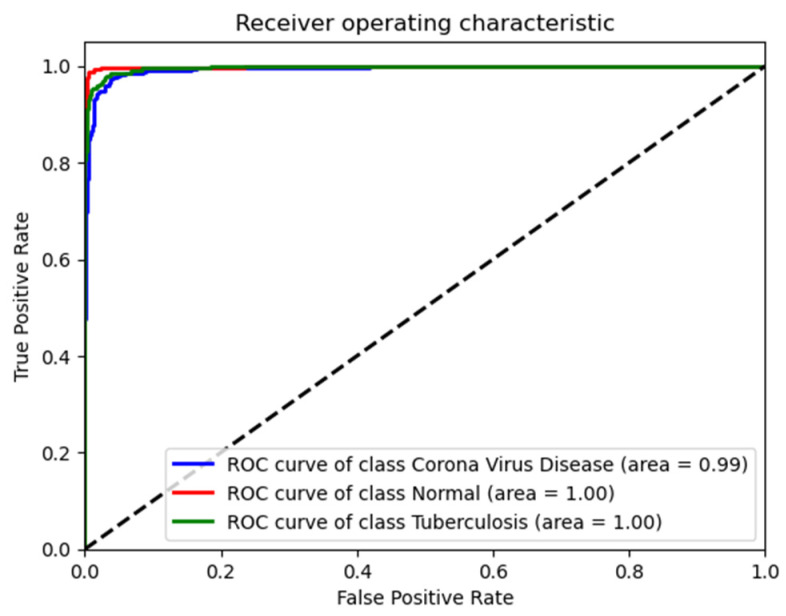
Receiver operating characteristic (ROC) curves for the ResNet50 model in diagnosing lung conditions, on validation dataset.

**Figure 5 diagnostics-14-01334-f005:**
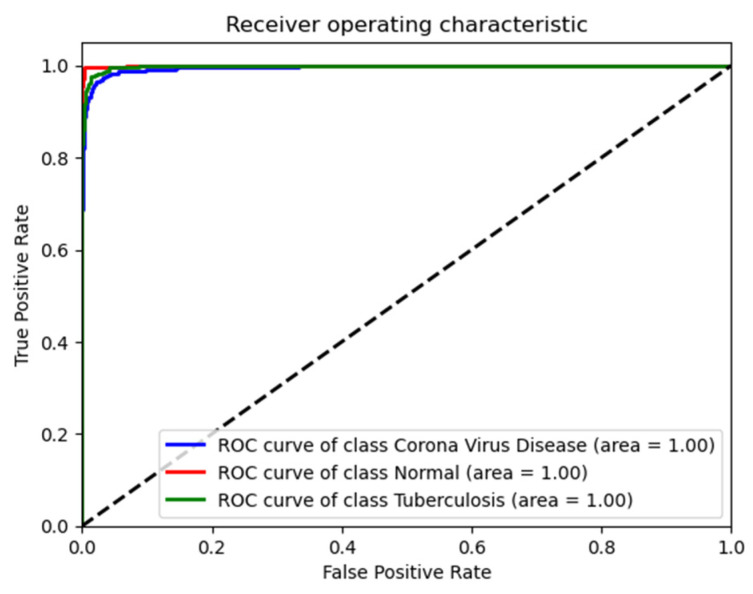
Receiver operating characteristic (ROC) curves for the ResNet50 model in diagnosing lung conditions, on test dataset.

**Figure 6 diagnostics-14-01334-f006:**
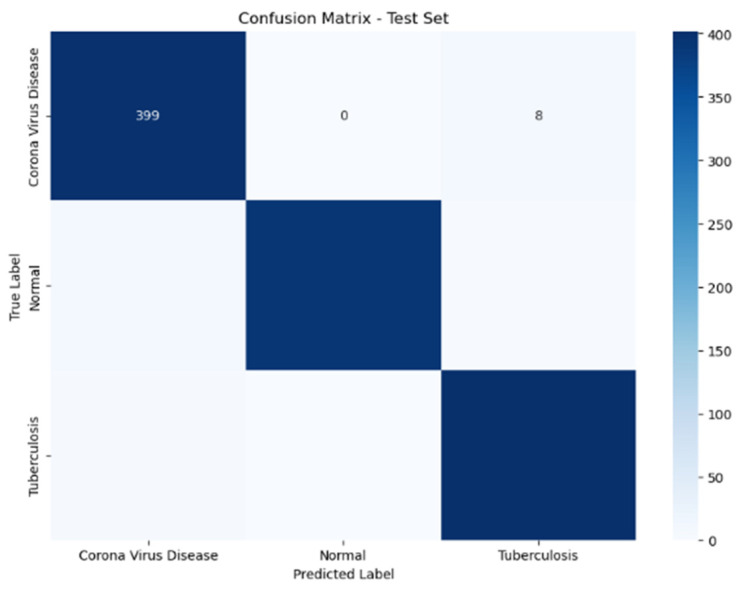
Confusion matrix of testing optimized using augmentation techniques.

**Figure 7 diagnostics-14-01334-f007:**
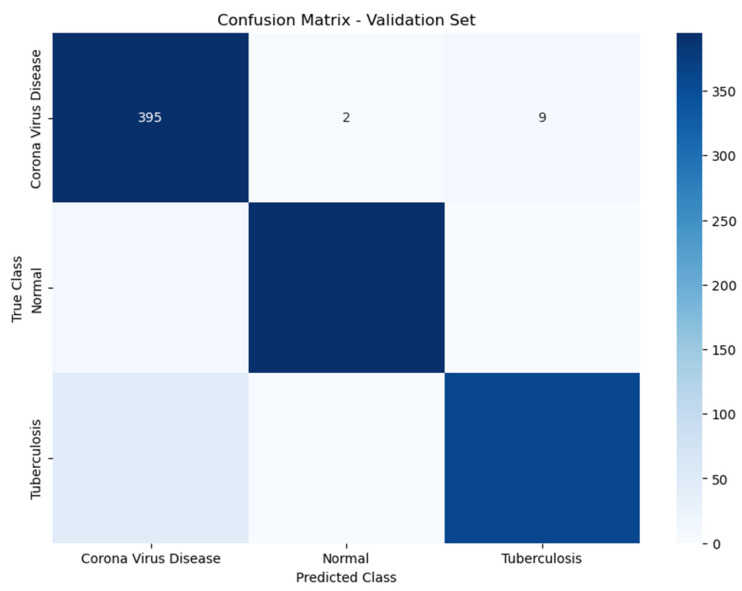
Confusion matrix of validation optimized using augmentation techniques.

**Figure 8 diagnostics-14-01334-f008:**
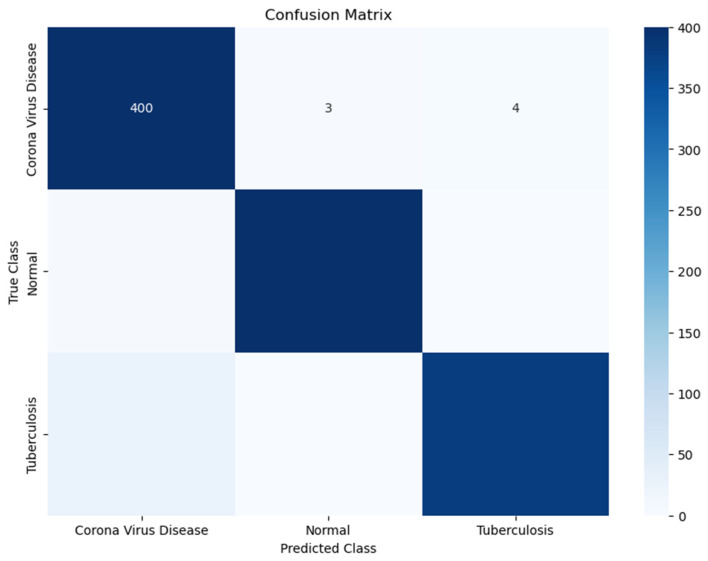
Confusion matrix of test dataset optimized using augmentation techniques.

**Figure 9 diagnostics-14-01334-f009:**
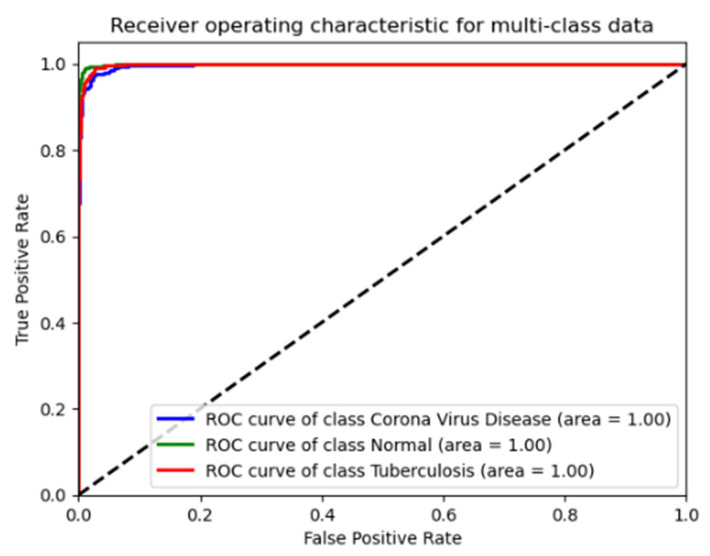
Receiver operating characteristic (ROC) curves for the VGG16 model in diagnosing lung conditions, on test dataset.

**Figure 10 diagnostics-14-01334-f010:**
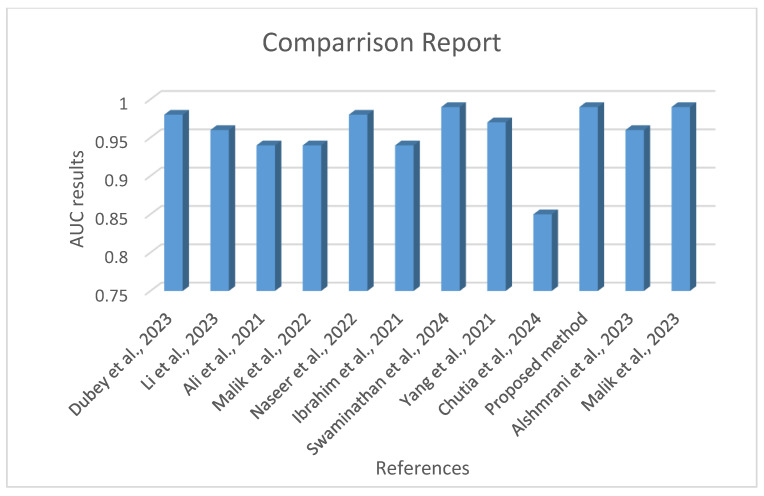
Comparison of AUC results for state-of-the-art lung disease detection models [9,10,11,12,13,14,15,16,17,18,19].

**Table 1 diagnostics-14-01334-t001:** Summary of Deep Learning Models Used for Lung Disease Diagnosis from Imaging Data.

Reference	Model Name	Methodology	Key Results
[9]	COVLIAS 1.0	Hybrid deep-learning model combining SegNet, VGG-SegNet, and ResNet-SegNet for lung segmentation.	AUCs ~0.96 to ~0.98, processing times <1 s, superior to NIH benchmarks.
[10]	ResNet50-Ensemble	Machine learning model for malignant–benign small pulmonary nodule classification on CT images.	High accuracy in classifying nodules, demonstrating potential for early diagnosis.
[11]	Decision Level Fusion	Deep feature selection with SVM and AdaBoostM2 for lung nodule classification.	Achieved 90.46 ± 0.25% accuracy on the LUNGx dataset.
[12]	BDCNet	Multi-classification CNN for COVID-19, pneumonia, and lung cancer diagnosis from chest radiographs.	Remarkable classification accuracy, enhancing early diagnosis.
[13]	AlexNet with SGD	Analysis of CNN architectures for lung cancer detection on LUNA16 datasets.	Highest validation accuracy and performance metrics among tested models.
[14]	VGG19+CNN	Deep-chest: Multi-classification model for diagnosing COVID-19, pneumonia, and lung cancer.	Exceptionally high accuracy and comprehensive performance metrics.
[15]	GAN-VGG16	GAN based segmentation and VGG16 for classification to predict lung cancer.	Predicted lung cancer with 97% accuracy, effective for early stages.
[16]	Eleven CNN Comparison	Comparative analysis of eleven neural networks for small datasets of lung images of COVID-19 patients.	Provided insights into model performance, aiding in optimal model selection.
[17]	Modified DenseNet	Classification of lung diseases using an attention-based modified DenseNet model.	High accuracy, providing crucial visual insights for radiological assessments.
[18]	Deep-CXR	Deep learning architecture for multi-class lung diseases classification using CXR images.	Achieved high classification accuracy across various lung diseases.
[19]	CDC_Net	Multi-classification CNN for detection of COVID-19, pneumothorax, pneumonia, lung cancer, and tuberculosis using chest X-rays.	Superior accuracy and robustness with an AUC of 0.9953.

**Table 2 diagnostics-14-01334-t002:** Dataset distribution of the images.

Diseases	Image Description	Image
Tuberculosis	Image Format: JPEG, Image Size: (512, 512) Image Mode: RGB	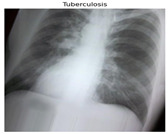
Normal	Image Format: JPEG, Image Size: (1580, 1410) and Image Mode: L	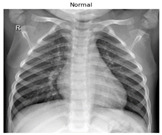
Corona Virus Disease	Image Format: JPEG, Image Size: (2000, 2000) and Image Mode: RGB	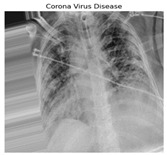

**Table 3 diagnostics-14-01334-t003:** Model Summary of the VGG16 with data preparation and Augmentation.

Layer (Type)	Output Shape	Param #
vgg16 (Functional)	(None, 7, 7, 512)	14,714,688
flatten_2 (Flatten)	(None, 25,088)	0
dense_4 (Dense)	(None, 256)	6,422,784
dropout_2 (Dropout)	(None, 256)	0
dense_5 (Dense)	(None, 3)	771
**Total params:** 33,985,355 (129.64 MB)		
**Trainable params:** 6,423,555 (24.50 MB)		
**Non-trainable params:** 14,714,688 (56.13 MB)		
**Optimizer params:** 12,847,112 (49.01 MB)		

**Table 4 diagnostics-14-01334-t004:** Model Summary of the ResNet50 with data preparation and Augmentation.

Layer (Type)	Output	Shape Param #
Conv2d-1	[−1, 64, 112, 112]	9408
BatchNorm2d-2	[−1, 64, 112, 112]	128
ReLU-3	[−1, 64, 112, 112]	0
MaxPool2d-4	[−1, 64, 56, 56]	0
Conv2d-5	[−1, 64, 56, 56]	4096
BatchNorm2d-6	[−1, 64, 56, 56]	128
ReLU-7	[−1, 64, 56, 56]	0
Conv2d-8	[−1, 64, 56, 56]	36,864
BatchNorm2d-9	[−1, 64, 56, 56]	128
ReLU-10	[−1, 64, 56, 56]	0
Conv2d-11	[−1, 256, 56, 56]	16,384
BatchNorm2d-12	[−1, 256, 56, 56]	512
Conv2d-13	[−1, 256, 56, 56]	16,384
BatchNorm2d-14	[−1, 256, 56, 56]	512
ReLU-15	[−1, 256, 56, 56]	0
Bottleneck-16	[−1, 256, 56, 56]	0
Conv2d-17	[−1, 64, 56, 56]	16,384
ReLU-171	[−1, 2048, 7, 7]	0
Bottleneck-172	[−1, 2048, 7, 7]	0
AdaptiveAvgPool2d-173	[−1, 2048, 1, 1]	0
Linear-174	[−1, 3]	6147
**Total params: 23,514,179**		
**Trainable params: 23,514,179**		
**Non-trainable params: 0**		
**Input size (MB): 0.57**		
**Forward/backward pass size (MB): 286.55**		
**Params size (MB): 89.70**		
**Estimated Total Size (MB): 376.82**		

**Table 5 diagnostics-14-01334-t005:** Validation and test dataset classification report of 3 conditions by ResNet50.

Validation Classification Report				
Class	Precision	Recall	F1-Score	Support
Coronavirus Disease	0.97	0.97	0.97	406
Normal	0.99	0.99	0.99	402
Tuberculosis	0.98	0.97	0.97	406
**Test Classification Report on**				
Coronavirus Disease	0.98	0.97	0.98	407
Normal	0.98	1.00	0.99	404
Tuberculosis	0.99	0.99	0.99	408

**Table 6 diagnostics-14-01334-t006:** Validation and test dataset classification report of 3 conditions by VGG16.

Validation Classification Report				
Class	Precision	Recall	F1-Score	Support
Coronavirus Disease	0.88	0.97	0.93	406
Normal	0.99	0.98	0.99	402
Tuberculosis	0.98	0.89	0.93	406
**Test Classification Report on**				
Coronavirus Disease	0.92	0.98	0.95	407
Normal	0.99	99	0.99	404
Tuberculosis	0.99	0.93	0.96	408

## Data Availability

Data available in a publicly accessible repository https://www.kaggle.com/datasets/omkarmanohardalvi/lungs-disease-dataset-4-types, accessed on 20 November 2023.

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
