# Peer review of "Augmenting Radiological Diagnostics with AI for Tuberculosis and COVID-19 Disease Detection: Deep Learning Detection of Chest Radiographs"

_diagnostics, 2024, doi:10.3390/diagnostics14131334_

Round 1

Reviewer 1 Report

Comments and Suggestions for Authors

1. There should be a brief introduction about tuberculosis, Covid-19, and X-ray imaging. I believe tuberculosis and Covid-19 are two different diseases infecting lung that is caused by different bacteria and virus.

2. Perhaps, the literature below could help to add the discussion on lung disease such as Covid-19, X-ray imaging, as well as deep learning.

WCS Low, JH Chuah, CATH Tee, S Anis, MA Shoaib, A Faisal, A Khalil, KW Lai, An overview of deep learning techniques on chest X-ray and CT scan identification of COVID-1, Computational and Mathematical Methods in Medicine 2021, 1-17, 2021

Comments on the Quality of English Language

-

Author Response

Greetings,

We appreciate the reviewers for their constructive comments, which have significantly improved the manuscript. After making the necessary corrections, the manuscript is now more enhanced and readable.

  With best Regards Manjur

Reviewer 2 Report

Comments and Suggestions for Authors

The article is devoted to a promising and very popular problem today - deep learning detection of chest radiographs (for Covid and tuberculosis). The article presented a very condensed overview of published research on this problem and a description of two learning models with which the authors worked.

However, this article is missing the main important part - results! It simply doesn't exist. A description of the proprietary models is presented both in the “methods” section and in the “discussion” section. Among the major concerns of this paper, I had comments on the presentation of information: in this form, this article will be difficult to understand for most readers, including radiologists.

For example, tables 2,3,4 and tables 3,6,7,8 are absolutely not clear for interpretation.

In my opinion, the article needs deep revision, a clearer and more understandable description of the methodology and the addition of original research results.

Author Response

Greetings,

We appreciate the reviewers for their constructive comments, which have significantly improved the manuscript. After making the necessary corrections, the manuscript is now more enhanced and readable.

  With best regards Manjur

Round 2

Reviewer 1 Report

Comments and Suggestions for Authors

I would suggest the titleof this manuscript be revised into "Augmenting Radiological Diagnostics with AI for Tuberculosis and Covid-19 Disease Detection: Deep Learning Detection of Chest Radiographs".

Comments on the Quality of English Language

-

Author Response

Greetings,

We would like to thank respected reviwers for their time and very insightful comments.

With best regards

Manjur

Reviewer 2 Report

Comments and Suggestions for Authors

The authors took into account the reviewer’s comments and concerns, significantly revised the article, and made important additions that significantly improve the understanding of the new learning models.

Author Response

Greetings,

We would like to thank each and every constrcutive comments from the reviewers. becasue of thier comments, the article is improved a lot.

With best regards

Manjur